# Reliability Assessment of On-Wafer AlGaN/GaN HEMTs: The Impact of Electric Field Stress on the Mean Time to Failure

**DOI:** 10.3390/mi14101833

**Published:** 2023-09-26

**Authors:** Surajit Chakraborty, Tae-Woo Kim

**Affiliations:** 1Department of Electrical, Electronic and Computer Engineering, University of Ulsan, Ulsan 44610, Republic of Korea; surajit5103@ulsan.ac.kr; 2Department of Electrical and Computer Engineering, Texas Tech University, Lubbock, TX 79409, USA

**Keywords:** AlGaN/GaN HEMTs, electric field, stress voltage, mean time to failure, channel temperature

## Abstract

We present the mean time to failure (MTTF) of on-wafer AlGaN/GaN HEMTs under two distinct electric field stress conditions. The channel temperature (*T_ch_*) of the devices exhibits variability contingent upon the stress voltage and power dissipation, thereby influencing the long-term reliability of the devices. The accuracy of the channel temperature assumes a pivotal role in MTTF determination, a parameter measured and simulated through TCAD Silvaco device simulation. Under low electric field stress, a gradual degradation of *I_DSS_* is noted, accompanied by a negative shift in threshold voltage (Δ*V_T_*) and a substantial increase in gate leakage current (*I_G_*). Conversely, the high electric field stress condition induces a sudden decrease in *I_DSS_* without any observed shift in threshold voltage. For the low and high electric field conditions, MTTF values of 360 h and 160 h, respectively, were determined for on-wafer AlGaN/GaN HEMTs.

## 1. Introduction

The AlGaN/GaN transistors exhibit favorable attributes for high-frequency power applications owing to their substantial band gap of 3.4 eV, exceptional breakdown field around 3.5 MV/cm, low on-state resistance, and effective heat dissipation capabilities [1,2,3,4,5]. In the absence of doping, the AlGaN/GaN heterostructure demonstrates a significant conduction band discontinuity. This, when coupled with the influences of piezoelectric polarization and spontaneous polarization, gives rise to the creation of a high-density two-dimensional electron gas (2-DEG) [6,7,8]. These features make AlGaN/GaN HEMTs uniquely suited for demanding applications that require high-power handling, high-frequency operation, and robust performance even in extreme conditions [9,10,11,12,13,14]. These advanced devices have emerged as key components in the field of electronics and power electronics, offering a wide range of benefits that make them indispensable in numerous cutting-edge applications [15,16,17,18]. AlGaN/GaN HEMTs have revolutionized power electronics. Their ability to handle high voltages and currents efficiently, along with their high-speed switching capabilities, has made them ideal for applications like power amplifiers, DC–DC converters, and power supplies [19,20,21,22]. They enable the development of compact and energy-efficient power electronic systems [23]. In the realm of radio frequency (RF) and microwave electronics, AlGaN/GaN HEMTs offer exceptional high-frequency performance, low noise characteristics, and high-power output [24,25,26,27]. These transistors are critical components in radar systems, wireless communication infrastructure, and satellite communications [28,29].

The wide bandgap confers substantial robustness against diverse manifestations of electrical overstress, encompassing direct current (DC), electrostatic discharge (ESD), and radio frequency (RF) stressors [30,31,32,33]. Hence, ensuring reliability emerges as a pivotal concern necessitating meticulous consideration throughout the developmental or material growth phases [34]. The mean time to failure (MTTF) serves as a critical parameter in assessing the longevity of devices within the context of long-term reliability. Subsequently, the mean time to failure (MTTF) can be extrapolated from the heightened test temperature to the standard operational temperature, typically hovering around 150 °C for gallium nitride (GaN) devices [7,35]. In terms of reliability categories, long-term reliability (around 1000 h according to JEDEC standard) at a three-temperature DC test is most used to determine device reliability [36]. Conducting measurements across various junction temperatures (at least three temperatures) facilitates the determination of activation energies (*E_a_*) through the application of the Arrhenius equation. Long-term high-power 50 V DC stress was induced on *L_g_* = 0.5 μm devices with an output current of 150 mA/mm (7.5 W/mm) for a duration of 816 h at channel temperature *T_ch_* = 280 °C, 300 °C, and 330 °C [37]. The initial drop in output drain current was observed at 24 h, and the period of stability was around 100–200 h. Beyond this point, the output current relative to time significantly decreased. After a comprehensive physical failure analysis, the emergence of crystallographic defects was ascertained within the entirety of the gate width in the AlGaN layer. This occurrence can be attributed to the manifestation of the inverse piezoelectric effect [38,39]. However, the analysis did not yield an estimation of the mean time to failure (MTTF).

The failure mechanism analysis of GaN-based HEMTs involves short-term reliability studies (<24 h), as conducted by various research groups [30,40]. Notably, hot-electron degradation has been well established in GaAs-based HEMTs, and similarly, the hot-electron effect remains a predominant degradation mechanism in GaN HEMTs. The aforementioned study investigated the hot-electron effect through DC short-term tests (<150 h) across diverse HEMT structures. The electroluminescence (EL) intensity exhibited a non-monotonic ‘bell-shaped’ trend when correlated with *V_GS_* while maintaining the *V_DS_* constant. Furthermore, a long-term accelerated test was conducted, spanning up to 3000 h, on a specific device at distinct bias points (*V_GS_* = 0 V, *V_DS_* = 6 V, (on state); *V_GS_* = −9 V, *V_DS_* = 32 V (off state); *V_GS_* = −4 V, *V_DS_* = 25 V (semi-on state)). Notably, under the semi-on state condition, a substantial degradation in transconductance (*g_m_*) was observed compared with the other conditions, indicating the presence of the hot-electron effect within the channel. In spite of a thorough examination of the degradation mechanism, the evaluation did not result in the computation of the mean time to failure (MTTF).

Numerous additional research groups have undertaken investigations involving three-temperature DC accelerated Arrhenius test aging, from which activation energies have been deduced [31,41,42]. High-temperature operating (HTO) tests were conducted by subjecting the devices to a consistent power dissipation of 6 W/mm. These tests were performed at varying channel temperatures of 204 °C, 232 °C, and 260 °C, all maintained under the same voltage condition (*V_DS_* = 25 V), over an approximate duration of 3000 h [43]. However, a comprehensive analysis of activation energy and MTTF was notably absent from the study.

Under a consistent voltage condition of *V_DS_* = 30 V, a high-temperature operating life (HTOL) test was executed for approximately 2000 h. This test encompassed three distinct channel temperatures: 210 °C, 225 °C, and 250 °C. The outcomes revealed a mean time to failure (MTTF) of 1.87 × 10^6^ h at a temperature of 200 °C, along with activation energy (*E_a_*) of 1.8 eV [44]. An accurate estimation of the channel temperature is of paramount importance for determining the precise mean time to failure (MTTF) values in GaN HEMTs. Employing a constant bias of *V_DS_* = 50 V and a power dissipation rate of 4 W/mm, devices were subjected to stress testing at three distinct base temperatures: *T_b_* = 160 °C, 175 °C, and 190 °C. However, the resulting MTTF values diverged based on the peak channel temperature (measured through Raman thermography) and the average temperature (measured via IR thermography). Specifically, two distinct MTTF values emerged: 10^9^ h and 10^6^ h [45].

Given the array of proposed stressors, degradation mechanisms, and associated degradation signatures, it is important to distinguish the precise stressors responsible for inducing particular effects. All prior investigations were carried out on packaged GaN HEMT devices. Limited long-term reliability studies exist on GaN epitaxial wafers or on-wafer devices [46]. In the current study, we investigated the extraction of activation energy and MTTF values under two distinct stress conditions, denoted as high and low electric field stress in on-wafer devices.

Assessing the reliability of gallium nitride high-electron-mobility transistors (GaN HEMTs) under various electric field stress conditions is crucial for several reasons: Understanding how GaN HEMTs behave under different electric field stress conditions allows for the optimization of their performance and operational lifetime [47,48]. By identifying stress conditions that may lead to degradation, manufacturers can develop strategies to mitigate these effects and design devices that operate more reliably and durably. As we mentioned previously, GaN HEMTs are often used in high-power, high-frequency, and critical applications such as aerospace, defense, telecommunications, and power electronics. In these applications, device failures can have serious consequences, including system downtime, mission failures, or costly repairs. Assessing reliability helps prevent unexpected failures and ensures the uninterrupted operation of these systems [49,50]. In some applications, GaN HEMTs are used in safety-critical systems, where their failure could pose significant risks to human safety or the environment [51,52,53,54,55,56]. Reliability assessments under different stress conditions help identify potential failure modes and enable the implementation of safety measures and redundancies to mitigate these risks.

The paper is structured as follows: Section 2 presents the materials and methods employed, Section 3 encompasses the results and subsequent discussion, and Section 4 encapsulates the concluding remarks.

## 2. Materials and Methods

The epitaxial layer structures were synthesized utilizing a low-pressure metal–organic chemical vapor deposition (MOCVD) technique on 3-inch sapphire wafers measuring 430 μm in thickness. This epitaxial configuration comprised an Al_0.25_Ga_0.75_N barrier layer (20 nm), a Ga-polarity GaN channel layer (150 nm), and a high-resistance GaN layer (2.4 μm) positioned atop the sapphire substrate. The schematic and process flow can be observed in Figure 1a,b, respectively. The device fabrication encompassed mesa isolation etching, the establishment of source–drain ohmic contacts, and gate patterning. Mesa isolation etching was executed employing a reactive ion etching (RIE) system. Following this, standard Ti/Al/Ni/Au (25/160/40/100 nm) metallization was applied to the source and drain regions to form ohmic contacts. These contacts were then subjected to rapid thermal annealing (RTA) at 830 °C for 30 s within an N_2_ environment, facilitating the formation of contacts on the AlGaN/GaN epi-structure. Metallization was achieved through the lift-off technique. Subsequently, photolithography was employed to pattern the Schottky gate contacts. The Schottky gate contacts, composed of Ni/Au (20/300 nm), were fabricated using e-beam evaporation. For surface passivation, an Al_2_O_3_ layer (3 nm) was deposited.

Figure 2 represents output characteristics of GaN HEMTs device of gate length, *L_g_* = 3 μm, source to drain distance, *L_sd_* = 7 μm and gate width, *W_g_* = 50 μm. The output characteristics show that at very high drain voltage (*V_DS_* > 20 V) with an increase of gate voltage from *V_GS_* = −1 V to 2 V leads to a decrease in output drain current (*I_DSS_*) because of self-heating effects [57]. To gain insights into the influence of temperature and characteristics on stress performance and electrical behavior, we conducted evaluations using a BA1500 and a 4155C semiconductor parameter analyzer (Keysight Technologies, Santa Rosa, CA, USA). These instruments were linked to a probe station (MS TECH 5500) (MSTECH, Gyeonggi, Republic of Korea) equipped with a temperature-controlled (Temptronic TP03000) (inTEST Thermal Solutions GmbH, Deutschland, Germany) heating plate, ensuring precise temperature control during the I–V (current–voltage) characteristic measurements. The experimental setup and characterization procedures of the devices are displayed in Figure 3.

### MTTF Determination Method

Given the fact that many applications demand device lifetimes spanning several years, accelerated-life tests serve as essential tools for efficiently gathering reliability data within a practical timeframe. Among the myriad factors used for accelerating degradation in electronic devices, temperature emerges as one of the foremost contributors, as substantiated by extensive historical evidence within the semiconductor industry. One common approach to modeling semiconductor device reliability is to use the Arrhenius model. This model is based on the assumption that the rate of device failures is exponentially related to temperature and can be expressed as follows:(1)$$\lambda =Ae^{\left(\frac{E_{a}}{kT}\right)}$$
where *λ* is the failure rate, *A* is a material constant, *E_a_* is the activation energy (a measure of the energy barrier for failure mechanisms), *k* is the Boltzmann constant, and *T* is the absolute temperature (in Kelvin).

This equation outlines the connection between temperature and the rate at which the device degrades due to a specific failure mechanism. The semiconductor industry has widely embraced this equation as a guiding principle for overseeing device operation under diverse temperature conditions. The Arrhenius model allows for the determination of an acceleration factor (*AF*), which relates the failure rate at elevated stress conditions (*T_stress_*) to the failure rate at normal operating conditions (*T_normal_*):(2)$$AF=e^{\left(\frac{E_{a}}{kT}\right)\left(\frac{1}{T_{normal}}-\frac{1}{T_{Stress}}\right)}$$

One crucial assumption in this methodology is that failure mechanisms are thermally activated, and the Arrhenius model accurately describes the relationship between temperature and failure rate. The accuracy of MTTF calculations relies on the validity of the acceleration factor and the assumption that failure mechanisms under accelerated testing conditions are representative of those under normal operating conditions. The channel temperature (*T_ch_*) of the device plays a vital role in determining the activation energy and acceleration factor. Temperature variations can significantly influence device reliability, so precise temperature measurements and control are essential during accelerated testing. Accurate measurement and control of channel temperature are critical, as temperature variations directly impact device reliability and influence the activation energy used in the model. The methodology also assumes that failure mechanisms are thermally activated and can be accelerated under stress conditions, making the calculated MTTF values relevant to real-world device performance.

In this study, we delineate two distinct stress zones, each characterized by specific combinations of high current and low electric field, as well as low current and high electric field. To comprehensively investigate these zones, we carefully selected specific bias conditions. Specifically, we opted for two distinct bias zones: one at a low voltage (*V_DS_* = 10 V) and another at a higher voltage (*V_DS_* = 25 V), each accompanied by power dissipation rates of 2 W/mm and 1.25 W/mm, respectively. These selected bias parameters are concisely summarized in Table 1. Additionally, we conducted experiments at three varying base temperatures: *T_b_* = 150 °C, 170 °C, and 190 °C. The determination of channel temperature for each bias condition is discussed in detail within the Results and Discussion section of this study.

## 3. Results

Figure 4a,b represent the bias stress condition of low electric field (*V_DS_* = 10 V and *V_GS_* = 1.3 V set for 200 mA/mm, power dissipation, P = 2 W/mm) and high electric field (*V_DS_* = 25 V and *V_GS_* = −1 V set for 50 mA/mm, power dissipation, P = 1.25 W/mm). Under the low electric field stress condition, the device operates in a fully on-state condition, and a conspicuous self-heating effect is evident in the output characteristics (Figure 2). Consequently, this scenario closely resembles a high-power state condition. Conversely, during the high electric field stress condition, the device is in an off state, resulting in a minimal self-heating effect. This aligns with a high-voltage state in the off-state mode. Figure 5a shows the electric field simulation of stress voltage *V_DS_* = 10 V and 25 V. A negligible electric field variation is evident inside the AlGaN barrier. Figure 5b illustrates the electric field simulation inside the GaN channel. At the gate edge to the drain side, the electric field increased 1.2 times higher at *V_DS_* = 25 V than at *V_DS_* = 10 V. As we mentioned above, stress condition *V_DS_* = 25 V, *V_GS_* = −1 V is in the off-state mode. Therefore, a negative voltage is applied to the gate of the GaN HEMT. This negative voltage creates a strong electric field that pushes electrons away from the channel region. The high electric field in the off state extends through the GaN material and depletes the 2DEG, preventing the flow of electrons in the channel. In the on-state condition (*V_DS_* = 10 V, *V_GS_* = 1.3 V), a less negative (or even positive) voltage is applied to the gate of the GaN HEMT. This reduces the electric field across the device. The reduced electric field allows the 2DEG to accumulate or populate near the interface between the GaN and AlGaN layers.

The channel temperatures were computed using the DC or electrical method as outlined in Reference [58] and are illustrated in Figure 6 for the device characterized by a gate length, *L_g_* = 3 μm; source-to-drain distance, *L_sd_* = 7 μm; and gate width, *W_g_* = 50 μm. Across a range of gate voltages, specifically from *V_GS_* = 0 V to 2 V, the disparity in channel temperature (*T_ch_*) remained negligible. For 2 W/mm and 1.25 W/mm power dissipation, channel temperature rise (*T_ch_*) was approximately 60 °C and 38 °C, respectively, from the base plate temperature (*T_b_*).

TCAD simulations were conducted to verify the channel temperature against the measurement data, as presented in Figure 7.

In the context of TCAD (technology computer-aided design) simulation, specific mesh settings were defined for precise modeling. The mesh width was established at 50 microns, with the primary spacing in the x-plane set at 0.25 μm for the source and drain metal regions. Similarly, the mesh spacing for the source-to-gate (*L_sg_*) and gate-to-drain (*L_gd_*) regions was set at 0.25 μm. In the y-plane, the meshing ranged from 0 to 0.50 μm, with a spacing of 0.1 μm. This area covered the “air” region (region number 1). Beyond that, the AlGaN barrier (region number 2) extended from 0.50 to 0.520 μm, with an aluminum composition of 0.25% and a mesh spacing of 0.01 μm. The GaN channel (region number 3) spanned from 0.520 to 0.670 μm, also with a mesh spacing of 0.01 μm. The buffer region (region number 4) ranged from 0.670 to 3.070 μm and was uniformly doped with carbon (p-type), maintaining a mesh spacing of 0.01 μm. The AlN nucleation layer (region number 5) was extremely thin, from 3.070 to 3.018 μm. Finally, the sapphire substrate (region number 6) was in the range from 3.180 μm to the end of the device. Three electrodes were defined as source (y·min = 0.40 μm, y·max = 0.65 μm), drain (y·min = 0.40 μm, y·max = 0.65 μm), and gate (y·min = 0.40 μm, y·max = 0.50 μm). The work functions for these electrodes were specified as 5.20 eV, 4.0 eV, and 4.0 eV for gate, source, and drain, respectively.

In the simulation process, the high-field mobility was computed utilizing the Farahmand modified Caughey–Thomas (FMCT) and GANSAT models, while the low-field mobility was determined using the Albrecht model. Various physical models, including Schottky–Read–Hall (SRH), Fermi–Dirac statistics (FLDMOB), CONMOB, Fermi, and KP, were considered in the model definition. The polarization parameter was set to 0.952.

To account for self-heating effects, a lattice temperature model (lat. temp) was incorporated for channel temperature estimation in TCAD modeling, where the substrate is stated as ‘’thermalcontact num = 1”, with the specific region defined as region number 5, external temperature (ext.temp) set as 300 K, and adjusted thermal resistance (*R_th_* = 1/α).

Additionally, the Selberherr impact ionization model (Impact selb) parameters, namely an1, an2, bn1, bn2, ap1, ap2, bp1, and bp2, were set to specific values, namely 2.9 × 10^8^, 2.9 × 10^8^, 3.4 × 10^7^, 3.4 × 10^7^, 2.9 × 10^8^, 2.9 × 10^8^, 3.4 × 10^7^, and 3.4 × 10^7^, respectively. These parameters are essential for accurately modeling the device’s behavior and performance in the simulation environment.

Figure 7a depicts the simulation results for a device under *V_GS_* = 0 V and *V_DS_* = 10 V conditions while maintaining a base plate temperature of *T_b_* = 300 K (27 °C). Notably, the highest channel temperature recorded was 327 K (54 °C) in close proximity to the gate edge.

Similarly, when the device was biased at *V_DS_* = 25 V with the same gate voltage, *V_GS_* = 0 V, the corresponding channel temperature escalated to 360 K (87 °C), as illustrated in Figure 7b. This change corresponds to an approximate temperature increase of 33 °C. Consequently, the temperature variation within the channel is contingent upon the stress voltage conditions. To determine the changes in channel temperature (*T_ch_*) resulting from fluctuations in drain currents, we conducted experiments to observe the behavior of drain currents under different temperature conditions. Our findings indicated a consistent linear decrease in drain current across various temperature settings [29]. Additionally, we computed power levels (*I_DS_ × V_DS_*) from the output characteristics of the device. Subsequently, we normalized the drain current data relative to different temperatures and power levels. These normalized values were used to construct graphs in Figure 8 (measurement data), representing the relationship between channel temperatures and power levels.

Notably, this exhibits a remarkable congruence between the TCAD simulation outcomes and our measurement data.

### 3.1. Low Electric Field with High Current Stress Experiment

Figure 9a presents the transfer characteristics (characterization at *V_DS_* = 10 V) at low electric field stress condition at *V_DS_* = 10 V and output current level maintained to *I_DS_* = 200 mA/mm for power dissipation of P = 2 W/mm. At a constant base plate temperature of *T_b_* = 150 °C the channel temperature was estimated as *T_ch_* = 215 °C. After 84 h of stress, *I_DS_* and *g_m_* dropped around 30 mA/mm and 18 mS/mm, respectively. At the same time, gate leakage current *I_G_* (defined at *V_GS_* = −10 V, *V_DS_* = 10 V) increased from 3.3 × 10^−4^ to 0.034 mA/mm, as shown in Figure 8b. The threshold voltage negatively shifted around Δ*V_T_* = −0.16 V. After 175 h of stress, *I_DS_* and *g_m_* decreased more around 38 mA/mm and 31 mS/mm, respectively, from the initial value (Figure 8). At the same point, the leakage current increased from the initial value of 3.3 × 10^−4^ to 0.051 mA/mm, and the threshold voltage shift was around Δ*V_T_* = −0.31 V from the initial value.

Figure 10a shows the output characteristics before and after stress of 84 h and 175 h. On-resistance (Ron) increased around ΔRon = 20 Ω·mm at *V_GS_* = 0 V after 84 h of stress, and no change was observed until 175 h. The failure time is defined at *I_DSS_* degradation up to 15%. All the degradation in the other two base plate temperatures (*T_b_* = 170 °C and 190 °C) are depicted in Table 2.

### 3.2. High Electric Field with Low Current Stress Experiment

Figure 11a shows the transfer characteristics at high electric field stress *V_DS_* = 25 V, *I_DS_* = 50 mA/mm, and the power dissipation set at 1.25 W/mm. After 36 h of stress, there seemed a slight increase in the output current from 387 mA/mm to 401 mA/mm at the base plate temperature of *T_b_* = 150 °C. The maximum transconductance (g_max_) also showed negligible change. But at the same time, the leakage current I_G_ increased from 9.12 × 10^−5^ mA/mm to 3.86 mA/mm, whereas no shift was observed in the threshold voltage (Δ*V_T_*), as shown in Figure 11b. After 62 h of stress, the output current (*I_DS_*) decreased around 84 mA/mm from its initial value, and *g_m_* also decreased from 337 mS/mm to 313 mS/mm (almost 24 mS/mm). However, no change was observed in the leakage current. Table 3 illustrates the degradation observed at the other two base plate temperatures, namely *T_b_* = 170 °C and 190 °C.

Figure 12a illustrates the output characteristics prior to and following stress periods of 36 h and 62 h. After 32 h of stress, *I_DSS_* exhibited an increase, but this trend reversed after 62 h of stress. Notably, the on-state resistance (R_on_) demonstrated an increase of approximately ΔR_on_ = 60 Ω·mm at *V_GS_* = 0 V after 32 h of stress, with no noticeable alteration observed until the 62 h stress point. The degradation observed at the other two base plate temperatures, i.e., *T_b_* = 208 °C and 228 °C, is depicted in Table 3.

Figure 13a demonstrates the degradation of *I_dss_* (which is defined at *V_DS_* = 5 V and *V_GS_* = 2 V) in three different channel temperatures calculated for the *V_DS_* = 10 V bias condition. No abrupt degradation behavior of *I_dss_* was observed in high temperatures. However, under high-stress voltage conditions (*V_DS_* = 25 V), the device’s stability was compromised, lasting no more than 15 h at *T_ch_* = 228 °C, as depicted in Figure 13b.

Figure 14 illustrates the *MTTF* values calculated for three different channel temperatures under specific voltage stress conditions. To calculate the activation energy, the well-known Arrhenius equation of mean time to failure (*MTTF*) can be expressed as follows [26]:(3)$$\begin{array}{l} MTTF=e^{-\left(\frac{E_{a}}{KT}\right)}\\ \ln[MTTF]=-\frac{E_{a}}{kT} \end{array}$$

Here, *MTTF* = mean time to failure; *k* = Boltzmann constant, 8.6173 × 10^−5^ eV K^−1^; and *E_a_* = activation energy (eV). From the slope of Equation (3), activation energy (*E_a_*) can be calculated.

Under the low electric field stress condition (*V_DS_* = 10 V), the calculated activation energy was *E_a_* = 0.32 eV, yielding an extrapolated lifetime MTTF = 360 h. Conversely, under the high electric field stress condition (*V_DS_* = 25 V), the estimated activation energy was *E_a_* = 0.54 eV, resulting in MTTF = 160 h. The possible degradation or failure at low electric field and high current stress is related to the diffusion process (*E_a_* = 0.32 eV). This diffusion can lead to the formation of conductive paths or short circuits within the device, increasing leakage current and reducing the breakdown voltage. For the high electric field and low current stress, this degradation is related to the hot-electron effect or electron trapping (*E_a_* = 0.54 eV) [59]. The obtained mean time to failure (MTTF) values for GaN high-electron-mobility transistors (HEMTs) are significant indicators of device reliability and can provide insights into their performance under different electric field stress conditions. In general, MTTF represents the expected time for a device to fail under specified conditions. It is a critical parameter for assessing device reliability. We calculated MTTF values for on-wafer GaN HEMTs under both low (*V_DS_* = 10 V) and high (*V_DS_* = 25 V) electric field stress conditions. These values indicate how long, on average, the devices can be expected to operate before a significant number of them fail. The lower MTTF under high electric field stress (160 h) suggests that the devices are more prone to failure when subjected to higher voltage stress, which is consistent with accelerated aging in high-stress conditions.

Our MTTF values were validated only for on-wafer/bare-wafer devices. The MTTF for on-wafer devices typically represents the reliability of the semiconductor material itself, without considering packaging and external factors. On the other hand, the MTTF for packaged devices takes into account not only the intrinsic reliability of the semiconductor material but also the effects of packaging, assembly, and the device’s operational environment. Packaged devices typically have a longer MTTF than bare wafers because their packaging contributes to their robustness and resilience. In summary, comparing the MTTF of a bare-wafer device with a packaged device is not a straightforward apples-to-apples comparison.

## 4. Conclusions

The presentation of MTTF data for on-wafer devices was contingent upon specific electric field conditions. The accurate determination of channel temperature assumes a critical role in the precise estimation of MTTF values. Furthermore, degradation parameters exhibited variations based on the specific stress voltage or electric field conditions. Moreover, when calculating MTTF for on-wafer devices, distinct electric field conditions yielded different values. These intricate details merit thorough consideration as they hold the potential to significantly enhance the long-term reliability of AlGaN/GaN HEMTs.

## Figures and Tables

**Figure 1 micromachines-14-01833-f001:**
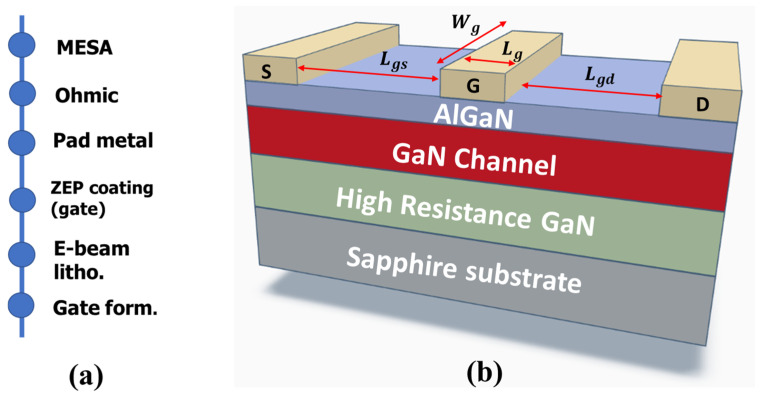
(**a**) Fabrication process flow; (**b**) schematic of AlGaN/GaN HEMTs.

**Figure 2 micromachines-14-01833-f002:**
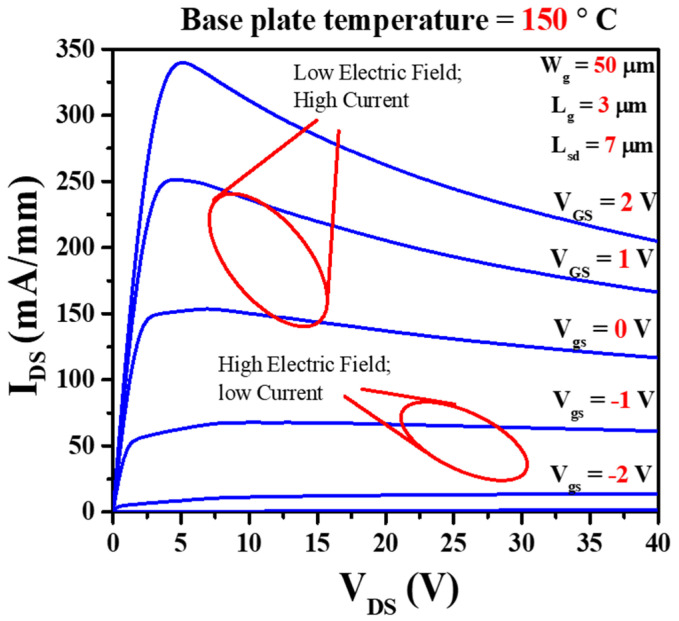
Output characteristics at base plate temperature 150 °C represent the stress zones of low electric field and high electric field.

**Figure 3 micromachines-14-01833-f003:**
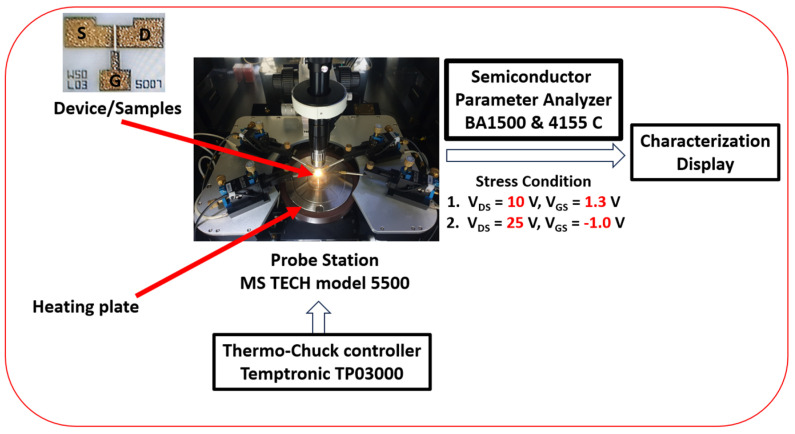
Experimental setup details and characterization procedures.

**Figure 4 micromachines-14-01833-f004:**
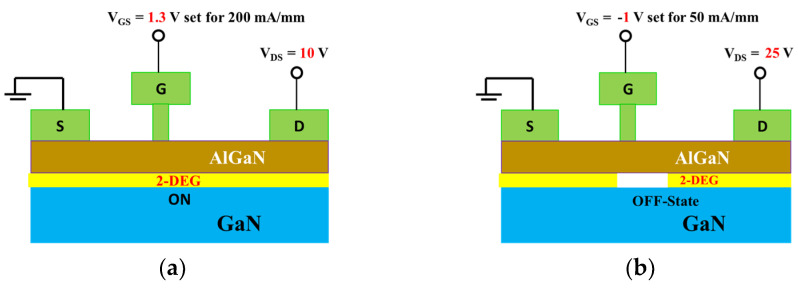
(**a**) Bias condition for low electric field and (**b**) high electric field region.

**Figure 5 micromachines-14-01833-f005:**
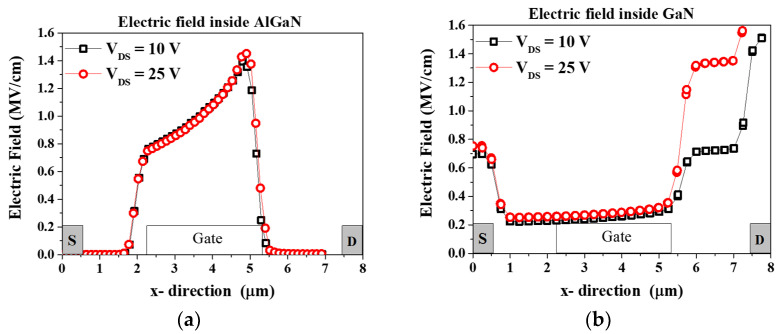
(**a**) Electric field simulation using Silvaco TCAD inside AlGaN barrier and (**b**) inside GaN channel.

**Figure 6 micromachines-14-01833-f006:**
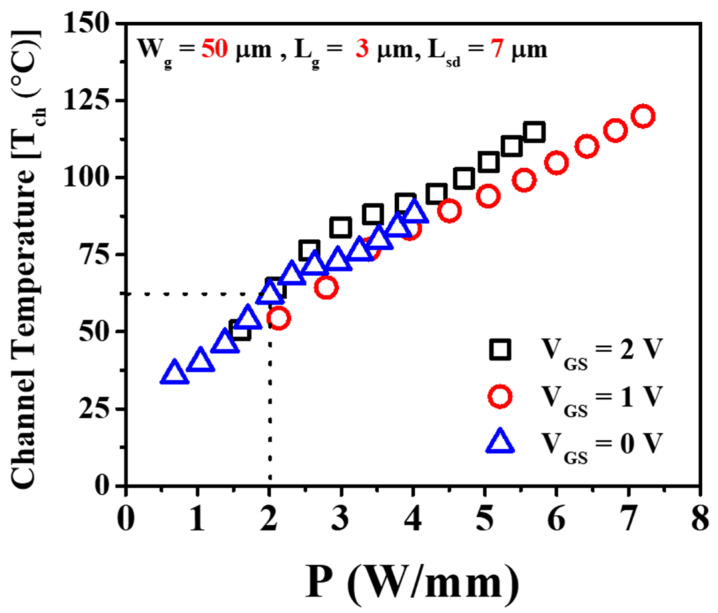
Channel temperature variation relative to the power dissipation of GaN HEMTs.

**Figure 7 micromachines-14-01833-f007:**
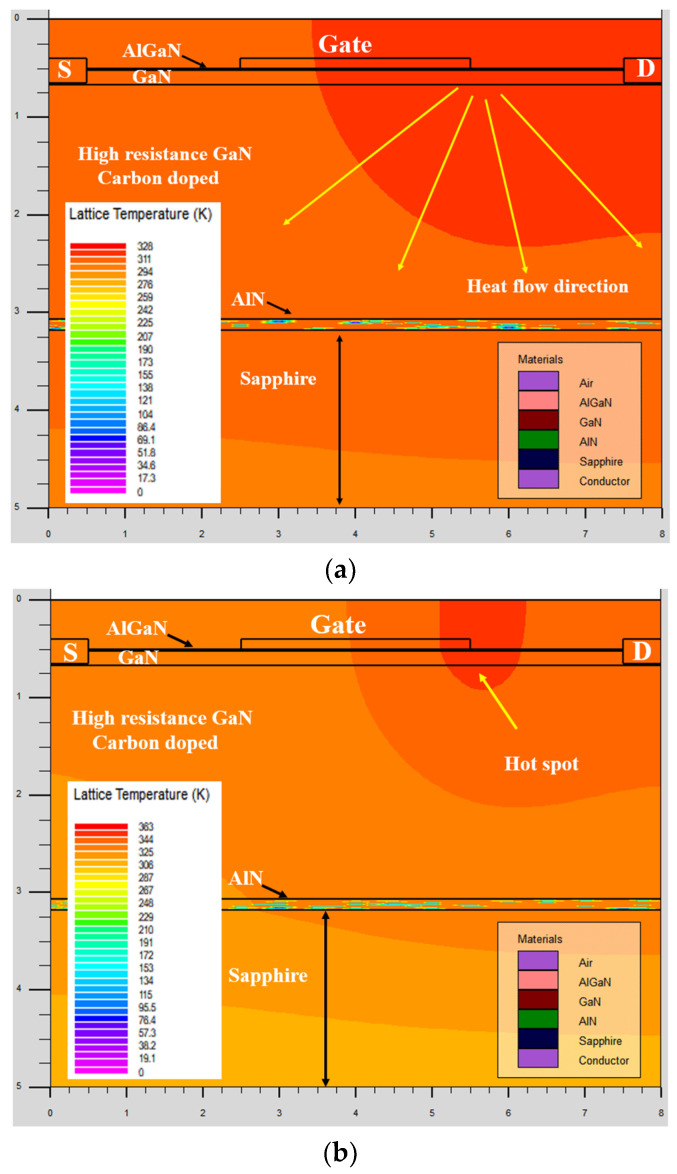
(**a**) TCAD simulation of the device at V_DS_ = 10 V and (**b**) V_DS_ = 25 V.

**Figure 8 micromachines-14-01833-f008:**
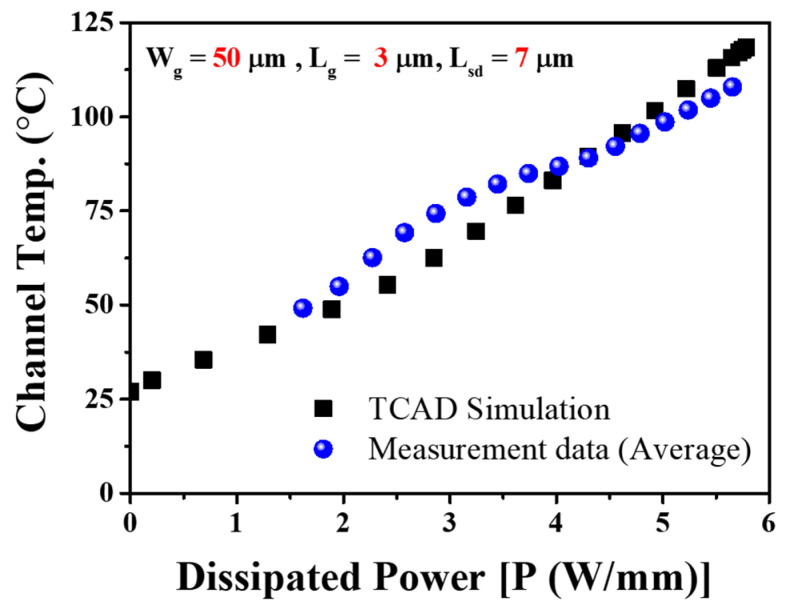
Measurement data (averaged from all three-gate voltages and TCAD simulation) show very close agreement to determine channel temperature.

**Figure 9 micromachines-14-01833-f009:**
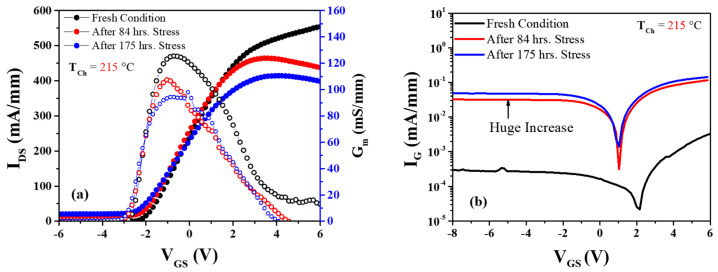
(**a**) Transfer characteristics after and before stress voltage *V_DS_* = 10 V; (**b**) Schottky characteristics depict gate leakage current after stress at the channel temperature, *T_ch_* = 215 °C.

**Figure 10 micromachines-14-01833-f010:**
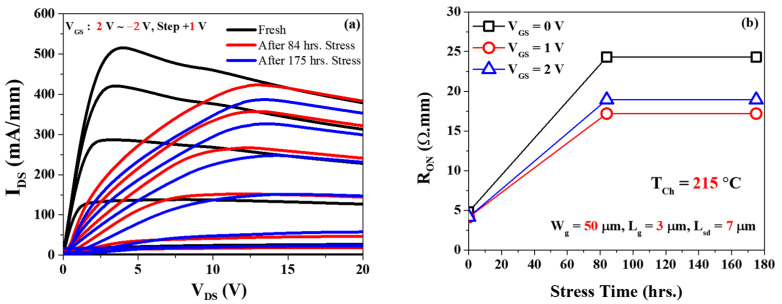
(**a**) Output characteristics after and before the stress of *V_DS_* = 10 V; (**b**) on-resistance characteristics after and before stress voltage at *V_DS_* = 10 V.

**Figure 11 micromachines-14-01833-f011:**
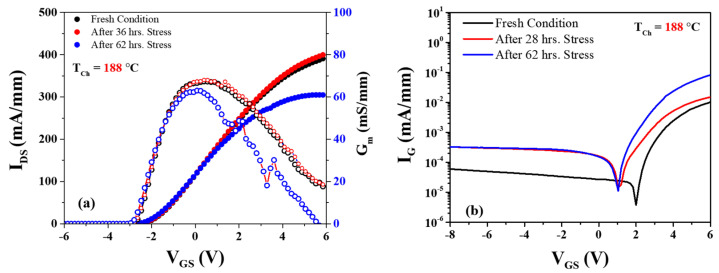
(**a**) Transfer characteristics after and before stress voltage *V_DS_* = 25 V; (**b**) Schottky characteristics depict gate leakage current after stress at the channel temperature 188 °C.

**Figure 12 micromachines-14-01833-f012:**
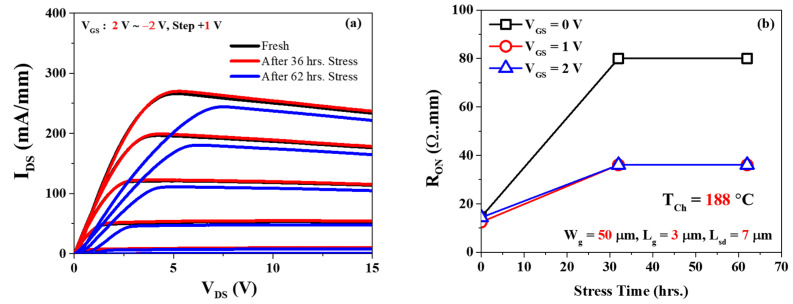
(**a**) Output characteristics before and after the stress of *V_DS_* = 25 V; (**b**) on-resistance characteristics before and after stress voltage at *V_DS_* = 25 V.

**Figure 13 micromachines-14-01833-f013:**
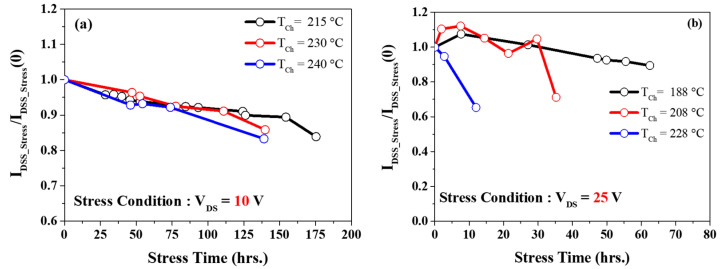
(**a**) *I_DSS_* degradation at low electric field stress voltage *V_DS_* = 10 V and (**b**) high electric stress voltage *V_DS_* = 25 V.

**Figure 14 micromachines-14-01833-f014:**
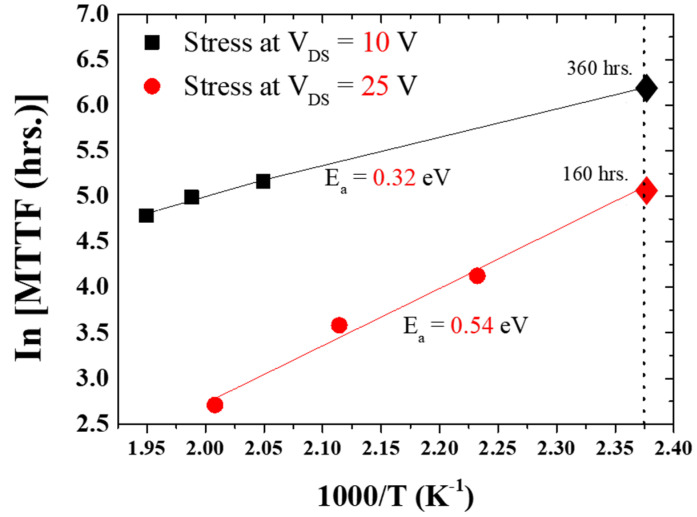
Mean time to failure (*MTTF*) analysis of two different electric field conditions.

**Table 1 micromachines-14-01833-t001:** Selected test condition for determination of MTTF values.

Sample Quantity	Stress Voltage, (*V_DS_* (V))	Current, (*I_DS_* (mA/mm))	Power, (P (W/mm))
5	10	200	2
5	25	50	1.25

**Table 2 micromachines-14-01833-t002:** Lifetime calculation at different base plate temperatures for low electric field stress.

Base Plate Temperature (*T_b_*) °C	Corresponding Channel Temperature (*T_ch_*) °C	Condition	Lifetime (h) (15% Degradation)
150	215	VDS = 10 V,	175
170	230	ID = 200 mA/mm	147
190	240	P = 2 W/mm	120

**Table 3 micromachines-14-01833-t003:** Lifetime calculation at different base plate temperatures for high electric field stress.

Base Plate Temperature (*T_b_*) °C	Corresponding Channel Temperature (*T_ch_*) °C	Condition	Lifetime (h) (15% Degradation)
150	188	V_DS_ = 25 V,	62
170	208	I_D_ = 50 mA/mm	36
190	228	P = 1.25 W/mm	15

## Data Availability

Data are available upon request from the corresponding author.

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
