# Peer review of "Reliability Assessment of On-Wafer AlGaN/GaN HEMTs: The Impact of Electric Field Stress on the Mean Time to Failure"

_micromachines, 2023, doi:10.3390/mi14101833_

Round 1

Reviewer 1 Report

In this paper, the Mean-Time-to-Failure (MTTF) of GaN HMET devices under different electric field stress conditions is studied by combining simulation and experimental results. This is an interesting and valuable research topic, which has important influence on the reliability of GaN devices. But there are still several issues worth discussing.

1. This paper describes that the AlGaN / GaN transistors have exceptional breakdown field exceeding 4 MV/cm, which is not the only or generally recognized reported result. The author needs to explain the basis for using this parameter.

2. The 2DEG of GaN HEMT device has a very small areal density, and then there is a self-heating effect. In this paper, the structure adopts Al0.25Ga0.75N barrier layer. Whether different Al components have a great influence on the self-heating effect, please give the reason for the selection of Al components.

3. In this paper, the channel temperature of the device is simulated based on SILVACO TCAD software, which physical models are used, and whether the lattice heating model can be explained in detail?

4. How to calculate the activation energy under low electric field stress condition ( VDS = 10V ) and high electric field stress condition ( VDS = 25V ) in Fig.13?

5. Why the stability of the device at Tch = 228°C under high stress voltage conditions ( VDS = 25 V ) is much worse than that at Tch = 208°C, as shown in Fig.12 (b).

6. There are some errors in the text format on page 11 of the article. Please check and modify.

7.         It is recommended to provide chip micrographs of the device and photographs of the device after packaging in the paper.

The English need to be improved, some expressions or sentences need to be corrected. such as: "The initial drop of output drain current observed until 24 hrs., period of stability around 100-200 hrs."

Author Response

Comment 1

Reviewer wrote:

  1. This paper describes that the AlGaN / GaN transistors have exceptional breakdown field exceeding 4 MV/cm, which is not the only or generally recognized reported result. The author needs to explain the basis for using this parameter.

Our response:

Thank you for pointing out this issue. We corrected this information.

Corresponding change in manuscript: Yes. Highlighted in yellow in the main manuscript.

Location of change: Section: 1 Introduction, Page- 1.  Line no 24.

Comment 2

Reviewer wrote:

  1. The 2DEG of GaN HEMT device has a very small areal density, and then there is a self-heating effect. In this paper, the structure adopts Al25Ga0.75N barrier layer. Whether different Al components have a great influence on the self-heating effect, please give the reason for the selection of Al components.

Our response:

Thank you for your question. But due to the thickness of AlxGa1-xN barrier layer, thermal resistance is negligible. Therefore, we did not consider it in channel temperature calculation. But in future we will try to check this issue.

Corresponding change in manuscript: No.

Comment 3

Reviewer wrote:

  1. In this paper, the channel temperature of the device is simulated based on SILVACO TCAD software, which physical models are used, and whether the lattice heating model can be explained in detail?

Our response:

Thank you for your valuable comment. We have added the information to the manuscript.

Corresponding change in manuscript: Yes. Highlighted in yellow in the main manuscript.

Location of change: Section: 3 Results, Page- 8.  Line no 240-271.

Comment 4

Reviewer wrote:

  1. How to calculate the activation energy under low electric field stress condition ( VDS = 10V ) and high electric field stress condition ( VDS = 25V ) in Fig.13?

Our response:

Thank you for your valuable comment. We have explained it in our revised manuscript.

Corresponding change in manuscript: Yes. Highlighted in yellow in the main manuscript.

Location of change:

          Section: 3. Results

         Sub-section:3.2 High Electric field with low current stress experiment.

         Page-13,  line no. 339-345.

Comment 5

Reviewer wrote:

  1. Why the stability of the device at Tch = 228°C under high stress voltage conditions ( VDS= 25 V ) is much worse than that at Tch = 208°C, as shown in Fig.12 (b).

Our response:

Thank you for your valuable comment. The reason is related to the effect of both high temperature and high voltage stress which maybe exceed the safe limit operation of the device. 

Corresponding change in manuscript: No.

Comment 6

Reviewer wrote:

  1. There are some errors in the text format on page 11 of the article. Please check and modify.

Our response:

Thank you for pointing out this issue. We corrected it in our revised manuscript.

Comment 7

Reviewer wrote:

  1. It is recommended to provide chip micrographs of the device and photographs of the device after packaging in the paper.

Our response:

Thank you for your valuable comment. Currently, we don’t have this picture. We will consider it in future.

Corresponding change in manuscript: No.

Author Response

Comment 1

Reviewer wrote:

  1. We noticed that the sample quantity was marked as five under both the two distinct stress conditions in Table 1, whereas the measurement results of only one sample were presented in the following content. Could the authors comment on the sample-to-sample variability?

Our response:

Thank you for your valuable comment. We have shown the best devices results. It will be lengthier and more complex if we show all the samples comparison and result.
In future, we will consider it.

Corresponding change in manuscript: No.

Comment 2

Reviewer wrote:

  1. In Page 5, line 147, the authors stated that the electric field increased 1.2 times higher at drain voltage of 25 V compared to 10 V. It seemed that the authors tried to emphasize this statement, and we suggested that more explanations should be added to demonstrate the significance of this result.

Our response:

Thank you for your valuable comment. We have given explanation regarding this issue

Corresponding change in manuscript: Yes. Highlighted in yellow in revised version.

Location of Change: Section 3, Results; Line : 216-224.

Comment 3

Reviewer wrote:

  1. In page 5, line 153, the authors stated that ‘Across a range of gate voltages, specifically from VGS = -2 V to 0 V…..’’ Nevertheless, the gate-source voltages marked in Fig.5 were 0 V to 2 V, and we suggested that this point should be checked carefully.

Our response:

Thank you for pointing out this issue. We have corrected this information.

Corresponding change in manuscript: Yes. Highlighted in yellow in revised version.

Location of Change: Section 3, Results; Line : 229-230.

Comment 4

Reviewer wrote:

  1. In page 6, line 166 and 169, the authors stated that the channel temperatures were 327 and 360K under drain voltage of 10 V and 25 V , respectively, whereas the lattice temperatures in Fig.6 may not be clear to identify the corresponding values. In addition, the highest temperature marked with red can be different under different drain voltages, and we suggested that it would be better to distinguish the above temperatures with different colors.

Our response:

Thank you for pointing out this issue. The lattice temperature (K) of color contours presented real time simulation data showed in transparent fashion. We changed the pattern in opaque fashion.

Corresponding change in manuscript: Yes. Highlighted in yellow in revised version.

Location of Change: Section 3, Results, Fig 6 (a) & (b).

Comment 5

Reviewer wrote:

  1. In 7, the authors presented the measurement results of channel temperature, and the computation method was presented in Ref [30]. Could the authors provide more details about the calculation process for better understanding of the readers? Besides the simulation parameters set in the TCAD should be added to this part.

Our response:

Thank you for pointing out this issue. We have added the information.

Corresponding change in manuscript: Yes. Highlighted in yellow in revised version.

Location of Change: Section 3, Results; Line 281-289.

Comment 6

Reviewer wrote:

  1. In 8(a) and 8(b), we noticed that compared with the fresh condition, the gate-source voltage corresponding to the lowest gate current presented the negative shift trend after long-term stress. It seemed that the Schottky junction degraded after the long-term stress. Could the authors provide more explanations about the physical mechanism?  

Our response:

Thank you for this nice question. We have mentioned it in sub-section 3.2. The activation energy found 0.32 eV after long term stress. Therefore,  the possible cause of degradation is related to metal diffusion. This diffusion can lead to the creation of conductive paths or short circuits within the device, increasing leakage current and reducing the breakdown voltage.

Corresponding change in manuscript: Yes. Highlighted in yellow in revised version.

Location of Change: sub-section 3.2, line 350-352.

Comment 7

Reviewer wrote:

  1. We noticed that the both the stress voltage and the power can be different for the selected two test conditions, and suggested that the experimental results may be affected by both the electric fields and temperatures. Could the authors analyze the effects of the above two factors separately.

Our response:

Thank you for pointing out this issue. We have presented the result of MTTF (mean-time-to -failure) in different stress conditions which clearly stated that if the stress condition is changed then the MTTF values must change. MTTF analysis combined both the effect of voltage and temperature. Analyzing both factors separately is totally different methodology need to use. We will consider it in future.

Corresponding change in manuscript: No.

Reviewer 3 Report

Overall, the manuscript titled "Reliability Assessment of On-Wafer AlGaN/GaN HEMTs: Impact of Electric Field Stress on Mean-Time-to-Failure" presents an interesting study on the reliability of AlGaN/GaN HEMTs under different electric field stress conditions. The research is valuable, but there are some areas that require attention and improvement to enhance the clarity and rigor of the paper. Here are specific comments for the betterment of the manuscript:

    • Provide a more comprehensive introduction to the importance of AlGaN/GaN HEMTs and their relevance in various applications.
    • Why is it important to assess the reliability of these devices under different electric field stress conditions?
    • Describe the experimental setup and conditions in detail, including the range of stress voltages and power dissipation levels used in the study.
    • Explain the TCAD Silvaco device simulation method in greater detail. What were the key parameters and assumptions used in the simulations?
    • Provide more information about the sample size and statistical analysis, especially when calculating MTTF values.
    • Compare your findings with existing literature on AlGaN/GaN HEMTs reliability, if applicable.
    • Explain the methodology and assumptions used to calculate the MTTF values in greater detail. Ensure that it is clear how the channel temperature influences MTTF.
    • Discuss the significance of the obtained MTTF values. How do they compare with industry standards or similar studies?
    • Consider including recent research papers and relevant studies to provide context and support for your work such as 10.1007/s42341-022-00391-y, 10.1007/s00339-020-3342-x, 10.1166/jno.2019.2558, etc.

minor

Author Response

Comment 1

Reviewer wrote:

  1. Provide a more comprehensive introduction to the importance of AlGaN/GaN HEMTs and their relevance in various applications.

Our response:

Thank you for your valuable comment. We have added more comprehensive introduction and applications in the introduction section.

Corresponding change in manuscript: Yes. Highlighted in yellow in revised version.

Location of change: Section: Introduction; line : 28-41.

Comment 2

Reviewer wrote:

  1. Why is it important to assess the reliability of these devices under different electric field stress conditions?

Our response:

Thank you for your valuable comment. We have added a more comprehensive introduction and applications in the introduction section.

Corresponding change in manuscript: Yes. Highlighted in yellow in revised version.

Location of change: Section: Introduction; line : 105-119.

Comment 3

Reviewer wrote:

  1. Describe the experimental setup and conditions in detail, including the range of stress voltages and power dissipation levels used in the study.

Our response:

Thank you for your valuable comment. We have added one graphical abstract for experimental setup.

Corresponding change in manuscript: Yes. Highlighted in yellow in revised version.

Location of change: Section: Materials and Method, figure 3, line : 154-155.

Comment 4

Reviewer wrote:

  1. Explain the TCAD Silvaco device simulation method in greater detail. What were the key parameters and assumptions used in the simulations.

Our response:

Thank you for your valuable comment. We have added more information of TCAD parameters.

Corresponding change in manuscript: Yes. Highlighted in yellow in revised version.

Location of change: Section: Results, Line: 243-270.

Comment 5

Reviewer wrote:

  1. Provide more information about the sample size and statistical analysis, especially when calculating MTTF values.

Our response:

Thank you for your valuable comment. We have already provided the information of sample size in section 2, materials and method section. Our wafer is 3-inch in size. Please kindly check the line 128 and the quantity of the sample we selected for stress was 5 samples each. Already provided the information in table 1.

For the statistical analysis, we will consider it in future.

Corresponding change in manuscript: No.

Comment 6

Reviewer wrote:

  1. Compare your findings with existing literature on AlGaN/GaN HEMTs reliability, if applicable

Our response:

Thank you for your valuable comment. But our device’s MTTF values cannot be compared with other existing literature because most of the literature emphasis about packaged device’s reliability but in our case, reliability is on-wafer device.

Corresponding change in manuscript:  No.

Comment 7

Reviewer wrote:

  1. Explain the methodology and assumptions used to calculate the MTTF values in greater detail. Ensure that it is clear how the channel temperature influences MTTF.

Our response:

Thank you for your valuable comment. We have added more information in section 2.1.

Corresponding change in manuscript: Yes. Highlighted in yellow in revised version.

Location of change: Section 2.1, MTTF determination method, Line : 156-188.

Comment 8

Reviewer wrote:

  1. Discuss the significance of the obtained MTTF values. How do they compare with industry standards or similar studies?

Our response:

Thank you for your valuable comment. We have added the information.

Corresponding change in manuscript: Yes. Highlighted in yellow in revised version.

Location of change: Section 3: Results; Line 310-328.

Comment 9

Reviewer wrote:

  1. Consider including recent research papers and relevant studies to provide context and support for your work such as 10.1007/s42341-022-00391-y, 10.1007/s00339-020-3342-x, 10.1166/jno.2019.2558, etc.

Our response:

Thank you for your valuable comment. We have added those important citation in our manuscripts.

Corresponding change in manuscript: Yes. Citation added no. 12, 13 and 14.

Round 2

Reviewer 1 Report

I think it is OK, and the editor office can decide whether to accept this paper.

Reviewer 2 Report

The authors have made detailed and convincing response to the comments. This manuscript has been improved and we suggest that this paper can be published as it is.